# Task-Motion Planning System for Socially Viable Service Robots Based on Object Manipulation

**DOI:** 10.3390/biomimetics9070436

**Published:** 2024-07-17

**Authors:** Jeongmin Jeon, Hong-ryul Jung, Nabih Pico, Tuan Luong, Hyungpil Moon

**Affiliations:** 1Department of Mechanical Engineering, Sungkyunkwan University, Suwon 16419, Republic of Korea; nicky707@skku.edu (J.J.); jung.hr.1206@skku.edu (H.-r.J.); npico@skku.edu (N.P.); dauluonganhtuan@skku.edu (T.L.); 2Facultad de Ingeniería en Electricidad y Computación, Escuela Superior Politécnica del Litoral, ESPOL, Campus Gustavo Galindo, Guayaquil 09-01-5863, Ecuador

**Keywords:** social robots, software architecture, task-motion planning, human-robot interactions

## Abstract

This paper presents a software architecture to implement a task-motion planning system that can improve human-robot interactions by including social behavior when social robots provide services related to object manipulation to users. The proposed system incorporates four main modules: knowledge reasoning, perception, task planning, and motion planning for autonomous service. This system adds constraints to the robot motions based on the recognition of the object affordance from the perception module and environment states from the knowledge reasoning module. Thus, the system performs task planning by adjusting the goal of the task to be performed, and motion planning based on the functional aspects of the object, enabling the robot to execute actions consistent with social behavior to respond to the user’s intent and the task environment. The system is verified through simulated experiments consisting of several object manipulation services such as handover and delivery. The results show that, by using the proposed system, the robot can provide different services depending on the situation, even if it performs the same tasks. In addition, the system demonstrates a modular structure that enables the expansion of the available services by defining additional actions and diverse planning modules.

## 1. Introduction

According to the results of the survey [1], from a user perspective, an investigation of a list of tasks required in home environments expected from service robots shows that object manipulation is the essential skill for real service robots. Therefore, if social cognitive skills are reflected in object manipulation-based services, robots can develop into more human-like and social robots that provide friendly services.

As the proportion of the aging population continues to increase [2], there is a growing demand for services that support households and individuals in their daily lives. Consequently, there has been an increasing need for service robots capable of performing such services. In response to these demands, the existing service robots have developed into human-friendly robots that can stimulate users’ emotions, not only performing human commands as a repetitive and simple task replacement, which can be confirmed through recent service robots’ technology trends [3] of various artificial intelligence technologies for human-robot interactions. Various human-friendly robots [4,5,6,7,8] have been developed, and the main feature is that they interact with humans. However, these robots mainly focus on the services of interaction through voice or video and mobile-based navigation. This paper aims to satisfy the needs of human-friendly service robots by developing a task-motion planning system that can provide object manipulation services that incorporate social behaviors. We define social behavior as the robot’s behavior that reflects interactions with people and their surroundings. Even when performing the task with the same goal, constraints on the robot’s actions may arise depending on the user’s intent or the state of the working environment, leading to variations in the composition of primitive actions. For example, when a robot manipulates a knife, it can be considered a social behavior to grip the handle when the robot is performing a task and grip the blade part when the robot hands it over to a person. In this context, we define the constraint that requires the robot to grasp the blade part to perform social behavior as a social constraint that reflects social behavior in motion.

Studies related to various object manipulation tasks have been studied, including task planning [9,10,11], motion planning [12,13], object recognition [14,15], human intention prediction [16,17], and knowledge reasoning [18,19]. Some of these studies have been combined into new areas of robotics research, and one of them, the task-motion planning system, has also been developed [20,21]. There are also a series of studies on the design of the system architecture to fully integrate the individual research mentioned above for the autonomous manipulation tasks of service robots [22]. The work in [23] introduces a hybrid message passing architecture for humanoid robots. This focuses on reducing the intermodular communication latency with respect to the performance of architectures such as interface size and memory usage and demonstrates performance improvements on the modules related to recognition and motion planning at high response rates. However, it does not include a module for human interaction and task planning. The research by [24] uses a system architecture applicable to different single- and multi-robot systems. This system shows a self-adaptive system that influences robot behavior based on the results of human gesture recognition. Since there is no task planning module, a global task must be decomposed into multiple subtasks by hand. The work in [25] demonstrates a three-layer architecture designed for service robots capable of dynamic interaction with humans in indoor environments. The system uses a knowledge base to infer surrounding information, and the combination of actions that the robot can execute enables the robot to perform tasks in various task domains. However, the action sequence in each task domain is determined by the finite-state machine in the top layer. The work in [26] shows a system architecture for performing mobile manipulation in an indoor environment. When the robot receives commands from the user, it estimates the task to be performed and the name and position of the object being manipulated through natural language processing. The robot then performs the task after the task and motion planning. However, no matter what the type of manipulated object is, the robot performs motion planning based on the primitive shape of an object. The authors mentioned above show how individual modules work with the system architecture but do not include all the components of task planning, motion planning, object recognition, and knowledge reasoning. Moreover, even if the robot recognizes human intentions and objects with a knowledge base, only the task plan is affected and the motion plan is not. There are also studies that consider object functionality and affordance for robot manipulation [27,28]. However, to the best of our knowledge, there are no studies on the development of a system that provides a service based on this, considering human-robot interactions.

This paper describes a software architecture that enables robots to perform social behaviors that can interact with users based on the object’s affordance and surrounding environment information in the object manipulation. Object manipulation is one of the important applications in robotics, and open-source libraries are already provided in various fields such as object recognition [28,29], knowledge reasoning [30], motion planning [31,32,33], and task planning [34]. Instead of building each application from scratch, our objective is to construct and integrate modules related to the perception, knowledge base, motion plan, and task plan based on existing applications. Our previous studies [35,36] present a method to integrate them and show the implementation of a task-motion system through the experiments of a simple object manipulation task by a service robot. This study aims to provide more friendly services for robots and proposes a system that incorporates methods of combining with knowledge reasoning results to reflect social behavior in robot motion. While the social behaviors covered in this paper fall far short of the socio-cultural behaviors assumed when discussing social robots [37], the proposed system developed in a Robot Operating System (ROS) can autonomously provide services related to object manipulation tasks to users through all the processes regardless of the robot platform, and it has the following novelties. Firstly, we implement a new system structure that integrates a perception module that can recognize the affordance of an object, along with task planning and motion planning modules that can utilize the functional aspects of the object to be manipulated. The system shows how additional constraints can be added to adjust the goal so that the action is performed in a manner that conforms to social behavior. Secondly, the proposed system shows that, even if the robot performs the same task, the robot can perform different compositions of primitive actions depending on the constraints imposed by the state of the surrounding environment. This is achieved through a knowledge reasoning module that infers the state of the surrounding environment and the user’s intentions, and a recognition module that recognizes the affordance of the manipulated object. Lastly, we show that our system can manage the robot’s task-performing status using real-time reasoning. This can make the robot automatically react to a change in the service environment.

The remainder of the paper describes the proposed system. In Section 2, we present the modules implemented and the overall task process. In Section 3, we verify the proposed system through experiments and describe the results. We conclude in Section 4.

## 2. Task and Motion Planning System Architecture

In this section, to provide object manipulation services with social behavior, we present a combined task-motion planning system. In this section, an overview of the system architecture is provided, showing how each implemented module communicates with other processes and transmits and receives data.

### 2.1. Overview

The system proposed in this study has a task manager, a system manager, an action library, a perception manager, and a behavior manager similar to the architecture proposed in our previous research [35,36], and in this paper a knowledge manager is modified to assign social constraints, which are constraints to reflect social behavior in motion. The relationship between the modules is shown in Figure 1.

In the case of the perception manager, it recognizes objects in the task environment and transfers the position and orientation of the objects. In addition, to perform social behaviors, the perception manager transfers affordances that include a functional aspect of the objects. The knowledge manager makes abstract inferences based on user commands, detected objects, and robot states. Additionally, the knowledge manager estimates the task goal from the user’s commands, including the states of the robot and its relationship to other objects in the workspace. In addition, using the inference results, social constraints are provided to actions to be performed by the robot to change the robot’s movement. The actions the robot can perform are modeled and described in the action library, and this information is used to plan in the task manager. The task manager performs task planning based on the Planning Domain Definition Language (PDDL) [38]. Using the action models described in the action library and states inferred from the knowledge manager, it automatically generates the required files and executes task planning. Motion planners that can generate motion trajectories for actions the robot is capable of performing are managed by the behavior manager. Motion planners receive not only the 6-dof pose of the objects to be manipulated but also affordances and social constraints as inputs. As a result, we implement motion planners that can generate different motions even if they perform the same action or manipulate the same object. The System Manager requests and receives the necessary information in a specific task state from the implemented modules. We define several task state and transition rules with the state machine so that the system can operate the robot to reach the desired goal state.

### 2.2. Functions of System Modules

#### 2.2.1. Perceptual Reasoning

The functionality of each component of the detected objects must be known to the robot to perform activities that require social behavior in the manipulation of objects. It means that the robot requires the affordances of the object in addition to knowing what type of object it is. The perception manager records the position, orientation, and affordances of objects in three dimensions and sends them to the system manager using information from the vision sensor. The 6D posture information for a detected object is shown in Figure 2a and is derived using the robot’s base coordinate. Each functional component of the objects is represented by a 3D bounding box that contains the relative pose determined by the object’s base coordinate. As shown in Figure 2b, objects can be divided into several functional parts related to manipulation. For example, objects that can hold a drink consist of the following affordances.

Contain: Storing liquids or objects (e.g., cups’ entrances);Grasp: Using a hand to enclose (e.g., tool handles);Wrap: Holding by hand (e.g., cup outside).

The tumbler and glass have contain affordance that can contain objects and wrap affordance that can be wrapped by hand, while the mugs have additional grasp affordance that can be held by hand. In this paper, for experiments through simulation in Section 3, the 6D pose information of the object to be manipulated and the object affordances are predefined in a hierarchical structure and are published through ROS topic in real time using the simulator API.

The knowledge manager is a module that performs knowledge inference based on perception results to provide current and goal states to the task planner and to impose social constraints on robot motion. The task manager requires predicates to plan tasks, which is one of the ways to express state information, and the knowledge manager generates these predicates with information about recognized objects. Figure 3 is an example of a knowledge inference process showing state estimation in a situation where a user commands a robot for a drink. The knowledge manager uses the sensor data to make an estimate of the robot’s task state in response to a request from the system manager. It creates an openedHand(right_hand) predicate indicating that the robot’s right gripper is opened, graspedBy(left_hand, obj_juice) predicate that means the juice box is held by the left gripper, and emptyHand(right_hand) predicate indicating that the right gripper is empty. The knowledge manager also estimates relationships between objects. If objects are within the robot hand’s workspace, the inWorkspace predicate is generated; if objects are interrupted, the obstruct predicate is created. After state inference, the predicates are delivered to the system manager in ROS message format.

#### 2.2.2. Symbolic Task Planning

The task manager determines the action sequence automatically using findings from the knowledge manager’s abstract inference results. Domain.pddl and problem.pddl are script files needed by the PDDL [38]-based task planner for task planning, and the task manager includes the domain generator and the problem generator to generate these. When the system manager delivers state information and action information, the problem generator generates problem.pddl from the states, and the domain generator generates domain.pddl from the action information defined in the action library module. The PDDL planner is used for task planning when the script files are generated by the domain and problem generators. We use a fast-downward algorithm [34] based on the state search method to create an action sequence. A compound action sequence is submitted to the behavior manager once the task manager successfully builds it.

In the system proposed in this paper, the action library can provide all the necessary action information in the task plan and the motion plan. After obtaining the compound action sequence, the action library converts it into a primitive action sequence for motion planning. In defining the action model, the action library divides the actions into compound actions and primitive actions, and defines the network structure in which primitive actions are composed of compound actions. In addition, the action information defined in the action library includes parameters necessary for the task planner and motion planner. Figure 4 illustrates examples definitions of compound action and primitive action in the action library. Each action is defined by applying the PDDL style and syntax to describe scripts to increase the convenience of editors and to easily apply the existing PDDL planner. Based on the action definitions in PDDL, we define the precondition required to perform each action and the effect that describes how the environment will change after the action is executed. In addition to syntax based on the PDDL format, primitives defines what actions the compound action consists of and constraints defines which motion planner the primitive action should be planned with, and these are added. A more detailed definition of the action library and network structures is applied to the methodology of our previous research [35]. Figure 5 illustrates an example of the decomposition of an action sequence for pick-and-place task. The task involves first relocating the juice that is blocking the way to grasp the milk. As a result of task planning, an action sequence consisting of compound actions is generated, following the red arrow. Next, to perform motion planning, the actions are decomposed according to the defined primitives, resulting in a primitive action sequence following the yellow arrow.

#### 2.2.3. Motion Planning with Social Constraints

The behavior manager manages modular motion planners to perform motions corresponding to primitive actions. The implementation of the behavior manager module adds a method to reflect social constraints in addition to the method in our previous work [35]. The behavior manager uses the particular behavior modules acting as motion planners to calculate a collision-free trajectory that robot can move along to execute the primitive action and checks to see if the execution is feasible after receiving the current action that needs to be executed and the parameters required in motion planning. Each module has different required and optional parameters for planning. The inputs of the modules for manipulation planning are the current and desired pose of the end-effector and 6D pose data of the objects and obstacles. The input information of the mobile path planning modules includes two-dimensional current coordinates and goal coordinates in the grid local map. We defined the data format as shown in Figure 6 for general use in motion planning, including input parameters and affordances of objects to be used in various motion planners. These data, defined as the ROS message, include not only the name of the action and the robot group to perform the action but also the position and object messages defined as lower-level messages, which correspond to the type of geometric variable of the primitive action defined in the action library. In this paper, motion planners corresponding to primitive actions in action library are implemented for experiments in Section 3 that provide services in various task domains, and some of the lists are as follows:

ApproachToObject behavior: It is a motion that moves the robot’s gripper into place prior to graspable affordance of an object. Our approach motion planner uses the URDF files, including robot information and 3D mesh files, such as STEP format of the target object to manipulate, as well as 6D pose data. When collision-free path between current pose and desired pose is calculated, multiple candidate goals are provided for robot’s end effector to approach in the frontal direction of the target object as shown in the Figure 7a. If the object’s graspable affordance is provided as an option, as shown in Figure 7b, goal poses are generated to approach affordance, and the path is generated.PourObject behavior: Pouring is the action of pouring a beverage from one container into another, which is necessary to perform a manipulation task, such as providing a beverage. There are studies that use force sensors or algorithms that determine the affordance of the object as part of planning the pour motion for a stable pour [39,40]. We implement a planner that can reflect the affordance of an object by motion planning that only takes into account the tilting behavior of the beverage without considering the flow of the beverage. When the pouring behavior module receives the bounding box and affordances of the target objects to be poured, the planner creates several candidate goal poses accessible to the container to be poured based on contain affordance coordinates and calculates a goal pose that allows pitching motion without collision as shown in Figure 7c.TransferObject behavior: The transferring motion is a motion that carries a holding object. The transferring behavior module receives the 3D bounding box of the objects, the 6D pose for the goal position to be delivered, and the object affordances as an optional input. If the motion path between the goal pose and the current pose of the end-effector holding the object is planned, the result is returned. If the object affordance is included as optional input, the motion path between the goal pose and the current pose based on the object affordance is planned so that a specific part of the object faces the target point, as shown in Figure 7d.OpenCloseHand behavior: The robot gripper motion planner creates a motion trajectory to hold or release objects. The object can be held just by closing the gripper joint since the approach_arm action is carried out first in accordance with the action sequence to take a position before grasping the object. Therefore, we use a predefined trajectory of gripper joints to open and close the robot gripper.MoveBase behavior: The mobile motion planner calculates motion for moving the robot mobile to a target pose. This behavior module calculates the path trajectory based on the RRT star planner by considering the approach direction to the goal pose.OpenDoor behavior: The door is a manipulation object that the service robot can often encounter in an indoor environment, and there are several studies [12,41] to calculate the motion for opening the door using the robot. We briefly implement a module to plan a motion to open a container with a door in which an object can be inserted for experiments in Section 3.2. We calculate the motion path to open the door from the current position of the robot to the axis of the hinge of the door for the container with the door opened to the left and right.

The knowledge manager helps robots to plan motions considering the human intentions and convenience. When the action sequence generated from the task plan is acquired, the relationship between actions and the relationship between individual action and manipulation objects are inferred, and additional constraints called social constraints are added to the action by the knowledge manager in the form of predicate. In this paper, since the implementation of the inference algorithm is not the goal, as shown in Table 1, the rules for adding social constraints according to the task were defined in advance. Considering the affordance of objects (e.g., giving a cup to the human in the direction of the cup handle), behavior performance attitude (e.g., delivering an object with two hands and putting the drink on the tray and transferring it politely), and human intention estimation (e.g., delivering an object to the position of the extended hand) are some example constraints. These constraints are transmitted to the behavior manager to add or change the input parameters. According to the task plan, a parameter is provided for the approach_arm action to grasp an object called a cup with the left arm. However, due to constraints, an additional wrap affordance parameter is added to ensure that the arm approaches the affordance in the motion planner. Alternatively, for the transfer_object action, due to constraints, the parameter values are modified to change the robot group performing the action from single arm to dual arms, or the goal position is changed from human to hand.

#### 2.2.4. Task Management

For the robot to complete the operation task, the system manager is in charge of managing the entire task execution process. The system manager receives data from the implemented modules and delivers them to the required modules. This section outlines the data flow that the system manager uses to request and process data based on task states.

##### Task States and Transition Rules

The system manager obtains data for the current task from various modules and transfers the state until it meets the user’s demand in accordance with the state transition model specified by the state machine. We showed in Section 3 that this state machine is applicable to task domains of any complexity and is designed to determine the order of operations between modules within the proposed system. Task states are divided into task performance and task preparation processes, and we have newly defined and added several states to reflect social constraints in addition to those defined in previous studies [35]. The predefined task state and transition rules are as shown in Figure 8, and the operation methods of the modules in each state are as follows.

Standby: In the Standby state, the system waits before or after performing the operation task until the next user’s command is received, and the ROS service receives a predefined command script as input information.GetStates: When the user orders the robot, commands are transferred to the knowledge manager from the system manager. Then, knowledge manager infers to determine the goal states for the command’s execution as well as the current states of the robot and task environment. Then, the system manager receives a response from the knowledge manager. If inference fails, it is determined that the intent of the user command is not grasped and task state moves to the Standby to obtain the user’s command.TaskPlan: When the goal state and current state predicates are obtained, the system manager forwards them to the task manager and requests a task plan. If the action sequence is successfully generated after the task plan, the task state moves to GetConstraints, and, if sequence is not planned, the state transitions to GetStates after it becomes TaskRetry state to infer the current state and try the task plan again. If the task plan retry continues, the system determines that the task is not in a situation where it can perform and the task state moves to Standby to receive the command again.GetConstraints: When the task plan is completed, the system manager receives a compound action sequence. In the GetConstraints state, the system manager requests the knowledge manager to impose social constraints on each compound action.DecodePlan: In this state, a compound action sequence derived from a task plan is decomposed into a series of primitive actions. The system manager requests action models from the action library to know which primitive actions each compound action is divided into, and social constraints assigned in the previous state are inherited as parameters for the divided primitive actions.GetCurrentAction: After the primitive action sequence is decoded, the robot takes out primitive actions to be performed in the sequence one by one. If there are no actions to be performed in the sequence after the executions, it is determined that the task is successful and transitions to the TaskSucceeded state and becomes a Standby to receive the next command.GetSymbolicValues: When the action to be performed is determined, the system manager brings up a model of this action from the action library. From the behavior model, the system manager extracts symbolic parameter values of the behavior and transition state to the GetMetricValues to embody them.GetMericValues: GetMericValues state is a state where the primitive action’s symbolic parameter values are transformed into geometric values. The system manager identifies where the values recognized by the perception manager correspond and updates the symbolic values.UpdateValues: When the metric values of the primitive action are assigned, the values are updated to reflect the social constraints obtained from the GetConstraints state. The social constraint is inherited from the compound action, and refer to Section 2.2.4 for the detailed process.CheckPrecondition: Before performing motion planning, when the action parameter values are all embodied, compare the conditions between the precondition of the action and the current states. If they do not match, it is considered that the environment has changed and the task state moves to TaskRetry.GetMotion: In this task state, the system manager sends the metric values to the behavior manager to generate the motion trajectory. If there is no possible motion, the robot state transitions to the task retry state and retries in the GetStates state.ExecuteAction: The system manager delivers the motion trajectory to the robot hardware controller. Then, robot moves along the motion trajectory. The task state transitions to CheckEffect state if the robot has completed its motion.CheckEffect: Similar to the CheckPrecondition state, the robot compares the effect of the action with the current state. If the comparison results are consistent, the system manager determines that the current action is successful. The robot then transitions to GetCurrentAction and repeats the actions in the sequence.TaskRetry: There are many reasons for task failure, such as instantaneous failure to recognize an object and not performing a task plan, or the robot cannot grasp the object because the position or posture is incorrectly recognized. This failure is identified in the GetMotion and CheckEffect states, and it then enters the TaskRetry state, prompting the task to be re-planned using knowledge inference or the objects to be recognized again. The operation fails if the robot fails more times than a predetermined number.

##### Data Flow in Task States

In order to automatically plan and perform motion of an action, the parameters must be provided metric values in the message data as shown in Figure 6 before the GetMotion state. This part explains how the system manager assigns metric values to the message data depending on changes in the robot task state defined in Section 2.2.4 with a task example. Figure 9 shows the data flow that the system manager requests from other modules to acquire a primitive action sequence, assuming that a robot is providing the service and in a situation where the person inputs the command “Give me a cup”. The system manager delivers the user’s command to the knowledge manager to receive the goal and the current robot states that the cup should be held by a person, and then delivers it to the task manager to obtain a action sequence that holds and delivers the cup. Then, in the GetConstraints state, the knowledge manager imposes social constraints on each compound action. In the example in Figure 9, the knowledge manager adds the wrap affordance condition of the cup to the hold_object for the handle of the cup facing the person, and adds (i) respectfully with both hands, (ii) to the extended hand, and (iii) with the handle of the cup facing the person conditions to the transfer_object. In the next task state, compound actions are decomposed into primitive actions, and imposed social constraints are inherited.

After a primitive action sequence is decomposed from compound action sequence, the primitive actions are popped one by one in the sequence. Figure 10 shows the process of allocating values to the behavior message for the approach_arm behavior obtained in Figure 9. The system manager first retrieves symbolic values and action parameters from the action library, as well as geometric parameters from the behavior manager. In the behavior message, symbolic values are assigned to the parameters corresponding to the action parameters and required parameters, and the static_object and dynamic_object that are not assigned are automatically assigned a list of objects recognized by the perception manager. Geometric values corresponding to parameters assigned symbolic values are retrieved by the system manager from the perception manager, and symbolic values are overwritten by the corresponding values in the position or object message. Since conditions for wrap affordance are provided to the current action in the previous GetConstraints state, wrap affordance of the cup, which is a target_object, is obtained from the perception manager, and the affordance message is added to the object message to consider the affordance in the motion planning state.

## 3. System Evaluation

### 3.1. System Environments

The simulations of the task-motion system proposed in this study are performed in a dynamic simulator called CoppeliaSim with a Vortex physical engine. Our simulation considers the dynamics of the robot and objects as well as the contact constraints between the objects, which means that proper motion planning and control algorithms must be used for successful robotic manipulation. We implemented the task planner module in the task manager using the fast-downward setting, and we used MoveIt to calculate the inverse kinematics for the motion plan in the behavior manager. ROS is used for the implementation of all the modules of the system. The robot used in the simulation experiments is equipped with a 6-dof dual arm manipulator, two 2-dof grippers, and a 2-dof waist joint, providing a maximum reach of approximately 50 cm. Furthermore, the mobile robot is equipped with four omnidirectional wheels, allowing it to navigate at a maximum speed of 0.8 m/s. In each experimental environment, the 3D object information is stored in the perception manager, and the robot’s position, pose, and affordance are determined in real time in the simulator. We defined compound and primitive actions that the robot can perform and the list can be found in the Appendix A.

### 3.2. Task Simulation Results

This study focuses on the performance verification of the proposed task-motion planning system with social behaviors and assumes that the goal state and the social constraints inferred by the knowledge manager in each task domain are derived as shown in Table 1. We also assume that the robot transfers the objects to the user and set the experiments on the tasks according to the social constraints to confirm the results of providing services with social behaviors.

Cases 1–4 are the results of the task that the cup is located on the table and a human requests a cup as shown in Figure 11 Then, the following primitive action sequence is obtained from the task planning: (i) open_hand (right_hand); (ii) approach_arm (right_hand, obj_mug, pos_right_hand, pos_mug); (iii) close_hand (right_hand); and (iv) transfer_object (right_hand, obj_mug, obj_person, pos_right_hand, pos_person).

In case 1, since there are no social constraints as shown in Table 1, the robot moves the arm to a position where it can grasp quickly without considering the affordance of the cup. Then, while the cup is not tilted, the robot creates a motion path to a position near the person and moves the arm.

Case 2 provides the social constraint of transferring objects with both hands so that the robot provides the objects politely, so the condition of setting the robot group with both hands is added to the transfer_object action. Then, in the motion planning stage, even for the arm not holding the cup, the motion trajectory to the current pose and the goal pose is calculated and transmitted to the arm controller so that both grippers move to the goal pose as shown in Figure 11c. In case 3, we set up a situation in which the human reached out to grasp the cup. Assuming that the human intentions are recognized, the knowledge manager can add a social constraint of transferring the cup to the extended hand position. As a result, the motion planner sets the goal pose of the transfer_object action as the position of the extended hand and the cup is transferred near the hand of the human as shown in Figure 11d. Case 4 shows the social constraint that the handle of the cup faces toward the human when transferring the cup to the human. The wrap affordance of the cup is considered for the compound action hold_object, and this social constraint is inherited for approach_arm when hold_object is decomposed into a primitive action. Therefore, the candidate directions to approach the robot gripper are limited to the wrap affordance where the robot grasps and the human can hold the grasp affordance of the cup. Additionally, considering the grasp affordance for the transfer_object action, the motion planner sets the start pose and goal pose based on the coordinate system of the grasp affordance and calculates the motion path so that the handle of the transferred cup is directed at the human.

Cases 5 and 6 in Figure 12 are the result of the task in which a human is ordered to drink in an environment where several drinks (hot coffee and cold juice) and cups (mug and glass) are placed on the table. In case 5, the knowledge manager defines the goal predicate that a cup should contain coffee as a goal state. The action sequence is then obtained as a result of task planning: (i) open_hand (right_hand); (ii) approach_arm (right_hand, obj_coffee, pos_right_hand, pos_coffee); (iii) close_hand (right_hand); and (iv) transfer_object (right_hand, obj_coffee, obj_person, pos_right_hand, pos_person). After that, in the GetConstraint state, the knowledge manager assigns a social constraint that sets the target object as the mug for the pour_object action so that the pouring motion planner calculates a motion plan for the mug and hot coffee is poured into the mug.

In case 6, the transferring object is changed from coffee to juice, unlike in case 5, and the inferred goal predicate from the knowledge manager is that the juice should be contained in the cup. In the GetConstraint state, the knowledge manager imposes a social constraint on cold juice to be placed in a glass. The experimental results of cases 5 and 6 are shown in Figure 12b and Figure 12c, respectively, and we can see that the robot pours the drink into the appropriate cup according to the drink it grasps.

In cases 7 and 8 in Figure 13, unlike the previous tabletop task domain, the drink to be delivered is inside the refrigerator, and the situation requires the robot to take the drink out of the refrigerator and deliver it to the person. We assume that the robot is aware of the presence of beverages in the refrigerator in advance because probabilistic reasoning is not considered in this paper. The difference between case 7 and case 8 is the presence of social constraints. In case 7, we also assume that the knowledge manager estimates the goal state that the milk should be held by the user without a social behavior, but, in case 8, the knowledge manager infers that a predicate that the milk should be placed on the tray should be added to the goal state and a social constraint to transfer the tray instead of the milk. The task plan for the request to ”Give me a milk” involves accessing the refrigerator (move_base), opening the door (open_door), reaching for the milk (approach_arm), grasping it (close_hand), moving to the person (move_base), and transferring the milk (transfer_object). However, as shown in Figure 12b, it is impossible to perform the task due to the interference of the juice when trying to grasp the target object. In the state machine, when following the rules, the state transitions to TaskRetry, and the task plan is executed again. As a result, an action to clear the obstacle (Figure 12c) is added, enabling the continuation of the tasks. As a result of the performance of the task, in case 7, which has no social constraints, the robot opens the refrigerator door and the milk is taken out and transferred directly to the user, as shown in Figure 13d. However, in case 8, the robot stacks the milk on the tray and transfers it to the user as shown in Figure 13f because of the social constraints.

To analyze the performance of our system, we performed each of the eight object manipulation tasks mentioned in Section 3.2 50 times within a simulated environment and described the results in Table 2. The results include the average task success rate for each case, as well as the execution times of each system module and the robot operation times. A task was considered to be successful if the state of the robot, after executing the final primitive action in the action sequence, included the goal state. The time of knowledge reasoning includes the goal and current state reasoning times. All the motion planning in the primitive action sequence is included in the motion planning time. In case 3, the motion planning time and total operation time are higher than in the other experiments. This seems to be because the motion planner did not perform the motion planning for both hands simultaneously but performed it first for the hand that held the object and then performed it for the other hand. Additionally, due to the failure of the motion planning due to the collision between the hand holding the object and the other hand, the planning time increased and the task success rate decreased.

## 4. Conclusions

The proposed system architecture integrates the necessary components for autonomous service provision, including a knowledge reasoner, perception, task planner, motion planner, and task-motion interface. This integration endows the system with several novel features. We implement a new system structure that integrates a perception module that can recognize the affordance of an object, along with task planning and motion planning modules that can utilize the functional aspects of the object to be manipulated. The recognized object from the perception module not only includes primitive shape information but also includes object affordances, and, since the motion planner can consider the affordances, the robot can perform motion planning based on the functionality of the object. The system shows that the robot can perform different combinations of actions or modified motions based on the constraints imposed by the state of the surrounding environment, even when performing tasks with the same goal. This capability enables the robot to perform services that reflect the human-robot interactions. Furthermore, the proposed system can manage the robot’s task-performing status using real-time reasoning. By using knowledge reasoning to update the current task state in real time and to plan tasks, the system can re-execute tasks even when unforeseen situations arise in a dynamic environment.

The following describes the general task process to provide a robot service. First, the knowledge manager determines the human intention from the provided command in order to create the goal states and current states with recognized data from the perception manager. The task manager uses the inferred results to obtain the order of actions to reach the goal state through symbolic planning. From the reasoning results of the knowledge manager, social constraints are provided regarding the action that the robot should perform, and the behavior manager plans a motion trajectory based on this. The action model described in the action library is used by the system manager to enable the symbolic task plan to be translated into geometric a plan that is capable of motion planning. This process can be applied regardless of the complexity of the task, and we experimented with simulations for several object manipulation environments and confirmed the system architecture in which the robot can provide different services according to the social constraints imposed differently based on the results from the knowledge manager for the same command.

The proposed system’s modular architecture, which integrates the existing open-source planning and perception algorithms, and its PDDL-syntax-based action library, are expected to enhance the system extensibility and usability. In this paper, we have defined actions solely related to robot pick-and-place tasks and conducted experiments in a limited number of environments. However, we believe that, by adding more behavior models to the action library and implementing corresponding motion planners, it will be possible to apply our approach to a broader range of task domains. In addition, we verified the system with only limited state information in the simulation environment. If we update the system with a knowledge module that can infer various situations, we also believe that it will be a system that is applicable to real environments.

## Figures and Tables

**Figure 1 biomimetics-09-00436-f001:**
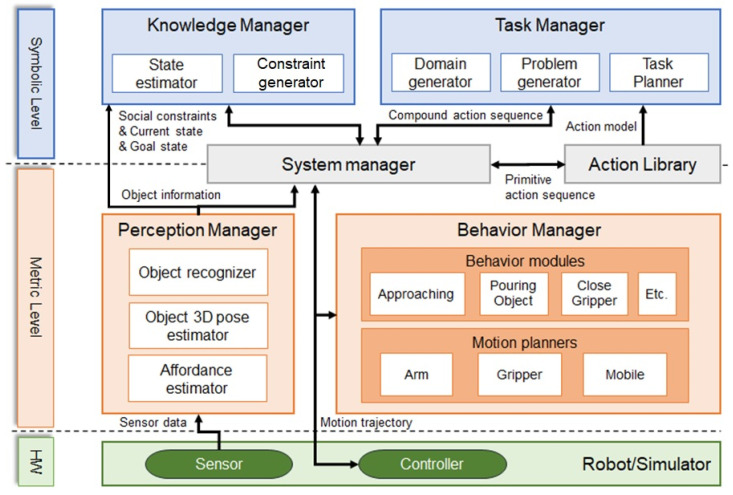
System architecture of the proposed task-motion planning system. Knowledge and system managers receive classified information from the perception manager. In order to execute the task in accordance with the task states, the system manager requests information from the action library, the task manager, and the behavior manager. All modules are implemented based on the ROS.

**Figure 2 biomimetics-09-00436-f002:**
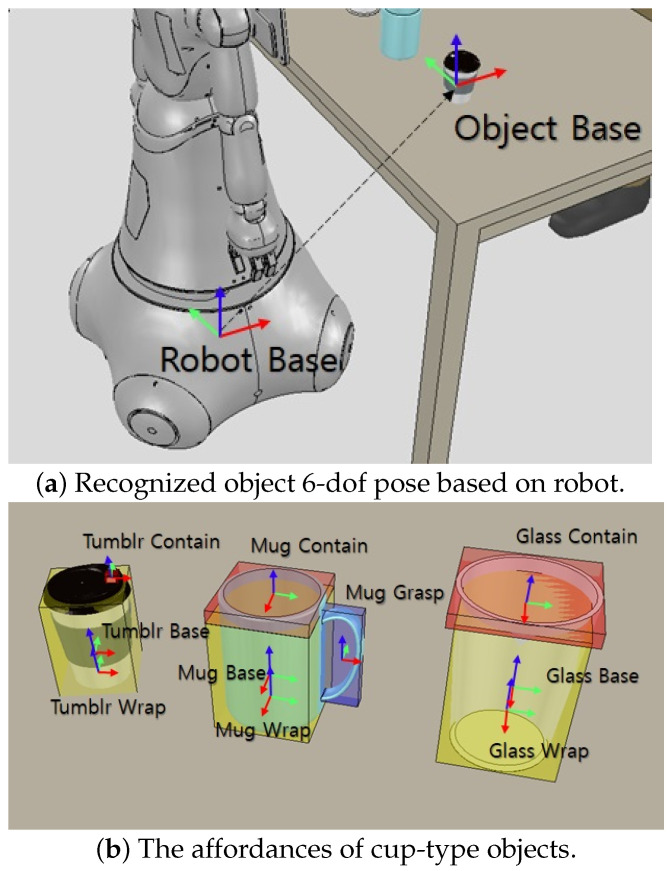
Object information used by the system. (**a**) 6-dof pose of object based on robot coordinates. (**b**) The object affordance defined as a bounding box in the simulation. The red bounding boxes are contain affordance, the blue boxes are grasp affordance, and yellow boxes are wrap affordance.

**Figure 3 biomimetics-09-00436-f003:**
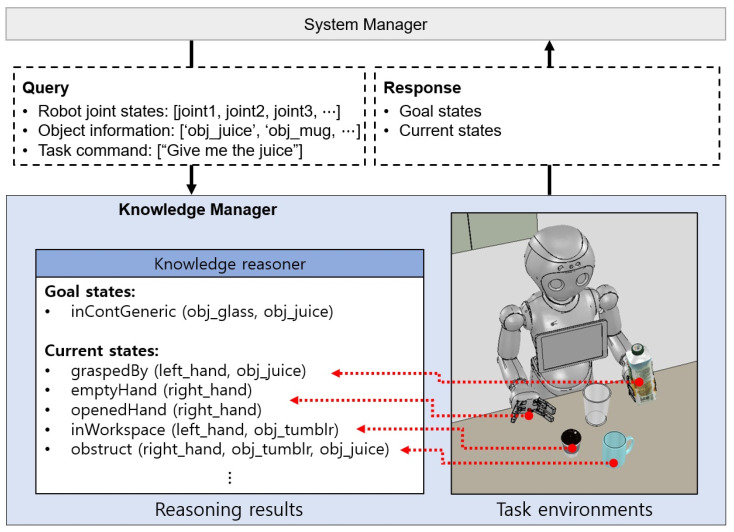
Example of the knowledge reasoning process results. When the system manager sends user commands, robot states, and recognition object information, the knowledge manager responds to the task goal and current states information in predicate form.

**Figure 4 biomimetics-09-00436-f004:**
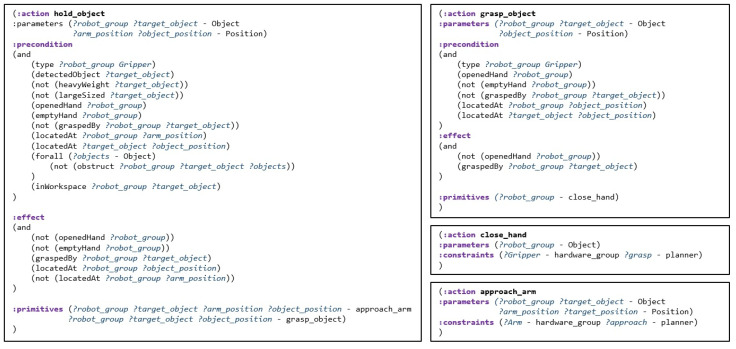
Example of actions described in PDDL format in the action library. An action of the left side is a compound action, which shows what primitive actions it consists of, and the right actions are the primitive actions.

**Figure 5 biomimetics-09-00436-f005:**
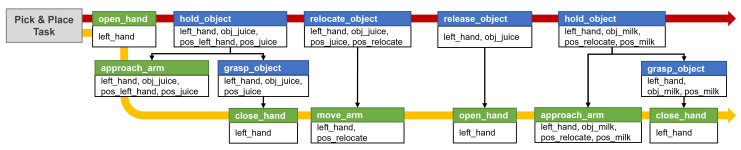
An example of action sequence decomposition. The names of compound and primitive actions are indicated by the blue and green boxes, respectively, while action parameters are indicated by the white box below. A primitive action sequence is depicted by the yellow arrow, whereas a complex action sequence is shown by the red arrow.

**Figure 6 biomimetics-09-00436-f006:**
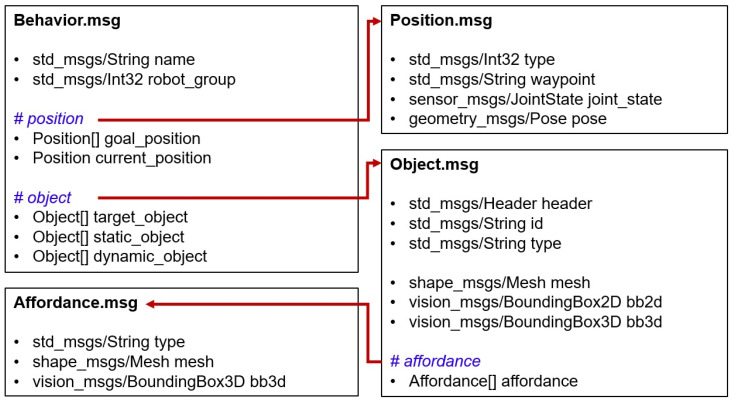
ROS message format implemented by behavior manager for use as input to the motion planner.

**Figure 7 biomimetics-09-00436-f007:**
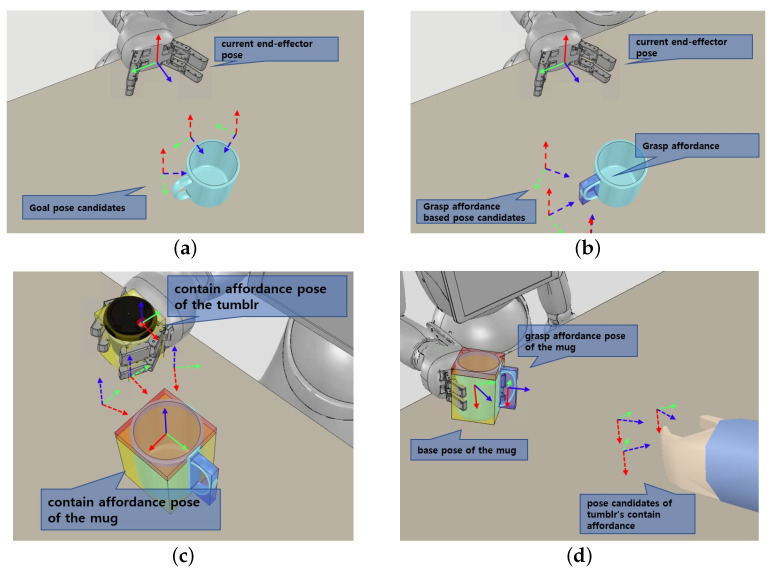
The initial and desired end-effector poses of behavior modules. (**a**) The object’s 6D pose depending on the robot base coordinates. (**b**) Predefined affordance of the objects within the simulation. The red bounding boxes are contain affordance, the blue boxes are grasp affordance, and yellow boxes are wrap affordance. (**c**) The pose candidates of the tumbler’s contain affordance for pouring the beverage into the mug. (**d**) The pose candidates of the grasp affordance ensure that the handle of the mug is oriented towards the user’s hand.

**Figure 8 biomimetics-09-00436-f008:**
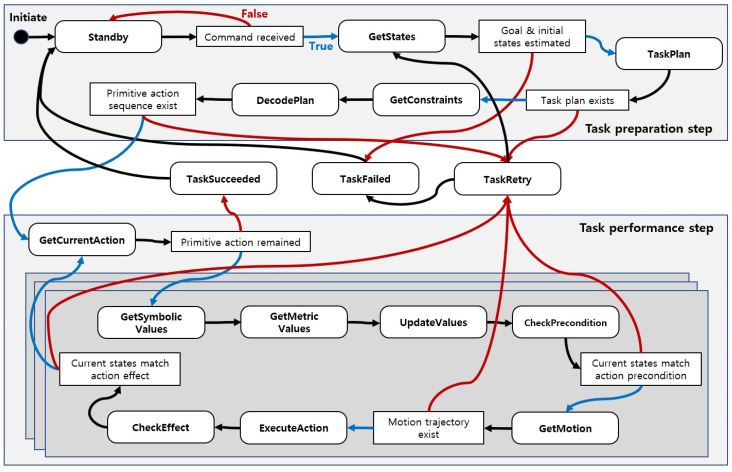
Task states and state transition rules. The white square box contains a conditional statement on the outcomes of the task in the previous state; if true, it moves in the direction of the blue arrow; if false, it moves in the direction of the red arrow. Regardless of how the prior assignment turned out, a black arrow indicates a state shift.

**Figure 9 biomimetics-09-00436-f009:**
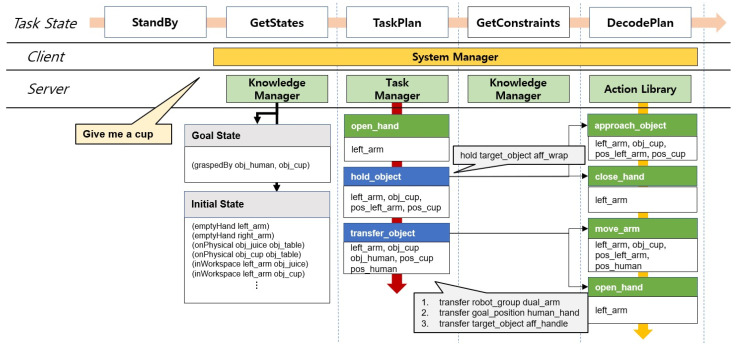
Data flow of the task preparation step.

**Figure 10 biomimetics-09-00436-f010:**
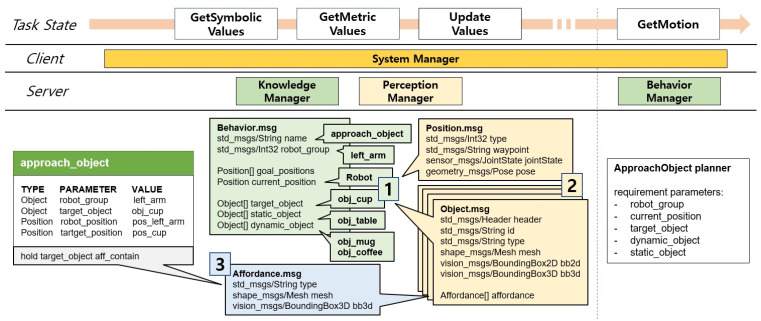
Data flow of the task performance step.

**Figure 11 biomimetics-09-00436-f011:**
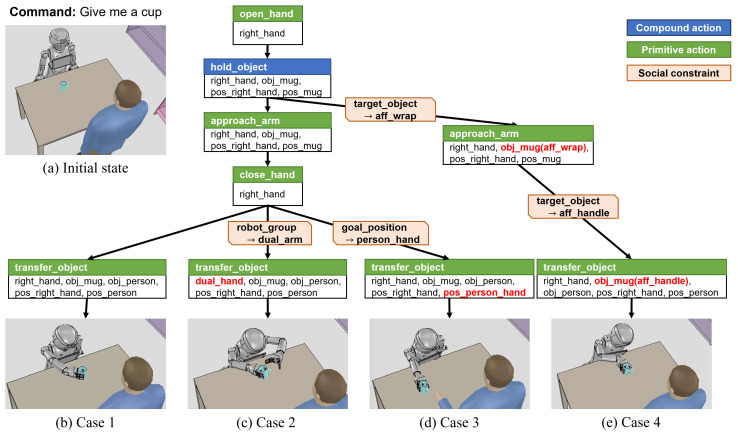
Experimental results of transferring a cup object. (**a**) Initial state when user asked for a cup. (**b**) Motion planning result without any constraints. (**c**–**e**) The results of performing task given a constraint. The red text denotes parameters changed by social constraints.

**Figure 12 biomimetics-09-00436-f012:**
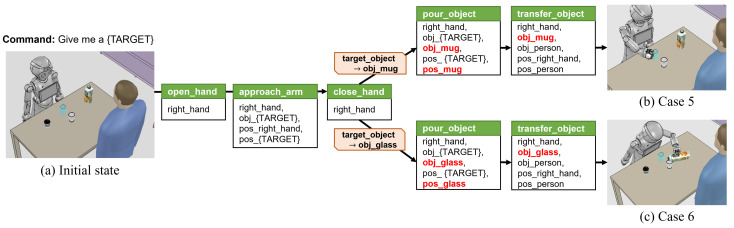
Experimental results of beverage pouring task. (**a**) It is assumed that there are hot coffee and cold juice box for drinks on the table, and a mug and glass to hold drinks. The coffee and juice correspond to targets case 5 and case 6, respectively. (**b**,**c**) Result of performing the task according to the requested beverage and given constraints.

**Figure 13 biomimetics-09-00436-f013:**
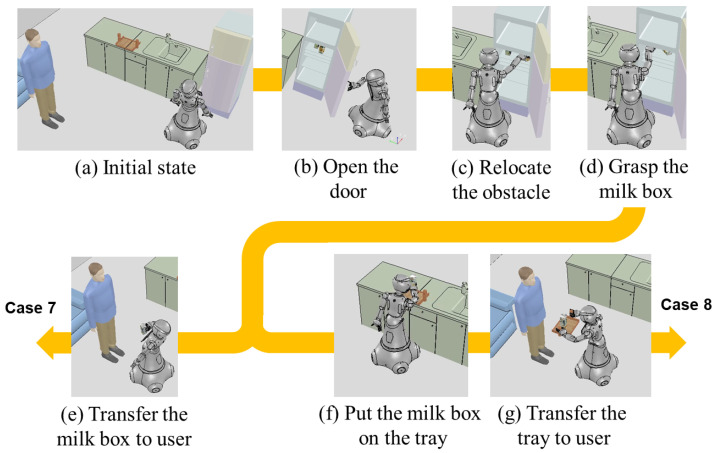
Taking out the drink from the refrigerator and delivering it to the human. (**a**) Initial state assuming that the robot knows that there are drinks in the refrigerator. (**b**–**g**) The action sequence obtained from the task plan result and the execution results. An action Relocate the obstacle is added to sequence as a result of re-planning because the obstacle blocked the target object.

**Table 1 biomimetics-09-00436-t001:** The predefined goal states and social constraints according to the experiments set up in Section 3.2.

Case	Command	Goal Predicates	Social Constraints
1	Give me a cup	graspedBy obj_human, obj_cup	-
2			hold_object target_object aff_wrap transfer_object target_object aff_handle
3			transfer_object robot_group dual_arm
4			transfer_object goal_position person_hand
5	Give me a coffee	inContGeneric obj_cup, obj_coffee	pour_object target_object obj_mug
6	Give me a juice	inContGeneric obj_cup, obj_juice	pour_object target_object obj_glass
7	Give me a milk	graspedBy obj_human, obj_milk	-
8		graspedBy obj_human, obj_milk belowOf obj_tray, obj_milk	transfer_object target_object, obj_tray

**Table 2 biomimetics-09-00436-t002:** Results of each experiment shown in Figure 11, Figure 12 and Figure 13 in the CoppeliaSim simulation performed 50 times.

Measurement		Case 1	Case 2	Case 3	Case 4	Case 5	Case 6	Case 7	Case 8
Success rate		100%	100%	86%	100%	92%	98%	98%	92%
Avg. execution time	Knowledge reasoning	0.08 s	0.08 s	0.05 s	0.37 s	1.52 s	1.81 s	1.62 s	1.38 s
	Task planning	0.32 s	0.43 s	0.28 s	0.25 s	0.47 s	0.31 s	0.49 s	0.51 s
	Motion planning	3.41 s	4.74 s	9.06 s	5.54 s	8.42 s	6.38 s	7.47 s	8.62 s
	Robot movement	34.42 s	31.38 s	57.45 s	32.6 s	24.19 s	13.78 s	245.12 s	320.89 s
	Total operation	38.23 s	36.63 s	66.84 s	38.76 s	34.60 s	22.28 s	254.7 s	331.4 s

## Data Availability

Data are contained within the article.

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
