# Peer review of "Task-Motion Planning System for Socially Viable Service Robots Based on Object Manipulation"

_biomimetics, 2024, doi:10.3390/biomimetics9070436_

Round 1
Reviewer 1 Report
Comments and Suggestions for Authors
This article is about robot software that performs the functions of a social assistant in everyday life.
The article focuses on the structure of software based on POS.
In general, the structure of the article is satisfactory. However, the scientific novelty of the study is unclear.
Notes on the work.
- The general task has not been formulated. The author focuses on the technical features of the software, while the actual analysis of the features of the software part is not carried out. The structure of the analyzer is formulated, but there is no comparison of technical characteristics. When presenting software, it is advisable to focus on this topic - performance, development speed, etc., assessing these parameters quantitatively.
- The set of commands that the robot operates is not specified. When examples are considered, some assumptions are made about functionality (for example, knowledge of the existence of the product). It is necessary to clearly describe for examples from which set of commands a solution is selected. How effective is the solution?
- The authors ignored optimal control problems and mathematical description. Perhaps this is not required here, but restrictions on the range of movements and speed of manipulation can be given.
- Figures 4,5, 8 contain names that are not explained in the text.
- A discussion section would be useful.
From my point of view, the article requires major revision; now it looks like the proceedings of a conference, and not a journal article. The authors communicate their approaches without giving the reader the opportunity to apply or evaluate them.
Author Response
We would like to thank editors and reviewers for the thoughtful review to our submission entitled "Task-Motion Planning System for Socially viable Service Robots based on Object Manipulation"
In this revision, we have addressed all the suggestions and comments from the reviewer. Changes in the revised manuscript were highlighted. We believe that the manuscript has been significantly improved and hope it has reached the standard of the journal
The answers to the reviewer's questions were written in pdf file and attached.

Reviewer 2 Report
Comments and Suggestions for Authors
Comments in the attached file.

Comments on the Quality of English LanguageAuthor Response
We would like to thank editors and reviewers for the thoughtful review to our submission entitled "Task-Motion Planning System for Socially viable Service Robots based on Object Manipulation"
In this revision, we have addressed all the suggestions and comments from the reviewer. Changes in the revised manuscript were highlighted. We believe that the manuscript has been significantly improved and hope it has reached the standard of the journal
The answers to the reviewer's questions were written in pdf file and attached.

Round 2
Reviewer 1 Report
Comments and Suggestions for Authors
The authors changed the text of the article based on my comments. The article has expanded significantly and become clearer. The article may be accepted